# ANM2026 Group Project Proposal

## Verifiable RCA: Root Cause Analysis with Temporal Evidence and Root-Centered Partial Causal Graphs

**Team Members:** Yisen Hong, Chenghao Li, Bowen Liang

**Date:** 2026/04/28

### 1. Introduction

Root cause analysis (RCA) for microservice systems seeks to identify the origin of failures from the many abnormal signals that appear during an incident. This remains difficult because a single fault can trigger multiple correlated changes across services, metrics, and telemetry sources, while many of those changes are only downstream effects. Although many RCA methods have been proposed, recent benchmark studies show that no single approach is robust across all scenarios, and that performance remains sensitive to the quality of anomaly evidence, structural assumptions, and dataset conditions [1][2][3].

This proposal is based on a simple idea: RCA should not stop at localization alone, and a partial causal graph should not be generated only after prediction as a descriptive artifact. Instead, the quality of a root-centered partial causal graph should contribute to how a root candidate is evaluated. A predicted root cause is more credible when it is supported by a partial causal graph showing how that candidate can account for the observed downstream abnormalities through temporally and structurally plausible propagation.

Accordingly, the project studies an RCA framework that couples root cause localization with partial causal graph support. The goal is to improve the robustness and trustworthiness of learned RCA by learning explicit temporal evidence, using service structure as a constrained prior, and evaluating candidate roots by both their local evidence and the quality of the root-centered partial causal graph they induce.

### 2. Motivation

From the practical needs, operators need more than a ranked list of suspicious services. In realistic incidents, many signals may become abnormal together, and a prediction is only useful if the user can check whether it plausibly explains the rest of the incident. This makes interpretability important, but not merely as post-hoc reporting. Instead, interpretability should come from a stronger requirement: the model should evaluate and present a root-centered partial causal graph grounded in the same evidence used for localization.

This proposal is motivated by that gap. Existing RCA methods already aim to identify true causes rather than symptoms, but current approaches still do not make causal support explicit enough in the prediction process itself.

### 3. Related Work

The most relevant prior work for this proposal falls into three groups.

**Statistical RCA methods.** These methods use handcrafted evidence such as z-score shifts, onset heuristics, persistence rules, and graph propagation priors. They are often interpretable and data-efficient, but the underlying evidence functions can be brittle across services and datasets [3].

**Learned RCA methods.** These methods learn temporal or multimodal representations and often improve ranking performance, but they typically optimize the final RCA label directly and leave intermediate evidence implicit. As a result, it can be difficult to verify why one candidate was selected over competing symptoms [5].

**Graph-based or causal-inspired RCA methods.** These methods motivate the use of topology and propagation structure, but recent evaluations show that causal-inference-based RCA is still not uniformly reliable across settings [2].

Taken together, these method families suggest a useful middle path: improve the quality and explicitness of temporal evidence, use structure to constrain causal plausibility, and generate a root-centered partial causal graph that makes predictions easier to trust.

## 4. Challenges

This proposal focuses on the challenges that are most directly related to that solution path.

**Ambiguous signals.** During an incident, several signals may become abnormal together, and the strongest anomaly is not always the true root-related signal.

**Implicit temporal evidence.** If a model only optimizes the final label, its internal signals may not correspond to meaningful RCA evidence such as anomaly strength, onset, persistence, or temporal order.

**Fine-grained ranking bias.** Metric-level prediction can inherit errors or miscalibration from higher-level service scoring, making it hard to recover the correct root-related label even when the correct service is already near the top.

**Weak verification.** Many methods use topology or propagation cues during ranking, but do not explicitly return a root-centered causal graph that allows the user to check how the predicted root explains the incident.

**Faithfulness of explanation.** Explanation is only useful if it is built from the same evidence that drives model decisions. Otherwise it becomes a post-hoc story rather than a trustworthy interpretation of the prediction.

## 5. Objectives

The overall objective of this project is to improve the robustness and verifiability of learned RCA by strengthening the temporal and structural evidence layer between telemetry and ranking.

More specifically, the project aims to:

1. learn explicit temporal evidence for each metric, such as anomaly strength, onset, persistence, magnitude, and related evidence signals;

2. use this evidence directly for service-level and metric-level root-cause ranking instead of relying only on latent end-to-end scoring;

3. incorporate service structure as a constrained prior so that ranking reflects propagation plausibility rather than anomaly magnitude alone;

4. generate a root-centered causal graph from the same model-used signals so that users can verify why the predicted root is plausible.

## 6. Proposed Methodology

The proposed methodology is centered on the joint use of temporal evidence and root-centered partial causal graphs for RCA. The core idea is that a root candidate should not be evaluated only by its local abnormality, but also by whether it can explain the observed downstream incident pattern through a temporally and structurally plausible partial causal graph.

### 6.1 Temporal Evidence Learning

The first component learns explicit temporal evidence from incident-centered telemetry windows. Instead of relying only on latent end-to-end representations, the model extracts signals that are directly useful for RCA, such as anomaly strength, onset, persistence, magnitude, and related temporal descriptors. These signals form the local evidence for each root candidate.

### 6.2 Root Candidate Evaluation with Partial Causal Graph Support

After local evidence is computed, each candidate root should also be evaluated through a root-centered partial causal graph. The graph should focus on the candidate root and show how downstream abnormalities can be explained through temporally and structurally consistent propagation. This component uses service structure as a constrained prior and builds the partial causal graph from the same evidence that supports ranking. The graph may include the candidate root service and metric-level label, supporting signals on the candidate root, downstream affected services and metrics, dependency-consistent propagation paths, and temporal ordering or lag relationships among affected signals. The key point is that the partial causal graph is not only an output artifact. It is used as a structured support object for root-cause evaluation. A candidate should be favored not only when it has strong local evidence, but also when it induces a partial causal graph that explains the observed downstream abnormalities well.

### 6.3 Faithful Explanation

After a root cause is selected, the same partial causal graph is presented as the explanation of the prediction. Because the graph is derived from the same evidence used in candidate evaluation, the explanation is faithful to the model's decision process rather than added afterward as a separate descriptive layer.

## 7. Dataset

The primary evaluation setting is RCAEval `RE2` [1], including:

`RE2-OB`, `RE2-SS` and `RE2-TT`

## 8. Evaluation Plan

The evaluation is intended to assess whether the proposed approach improves both root-cause localization and causal verification in the targeted benchmark setting.

### 8.1 Primary Metrics

- service `Acc@1`, `Acc@3`, `Acc@5`

- metric-level label `Acc@1`, `Acc@3`, `Acc@5`

### 8.2 Verification-Oriented Assessment

In addition to localization metrics, the method should be evaluated on the quality of its root-centered causal graph. Depending on what can be measured reliably, this may include:

- whether the graph covers observed downstream abnormal services

- whether the graph respects dependency structure

- whether temporal ordering in the graph is consistent with observed incident timing

- whether masking graph-supported evidence changes the predicted root score

## 9. Progress So Far

Initial experiments and benchmarking infrastructure are already in place.

## 10. Next Steps

- strengthen temporal evidence learning for localization;

- design and implement the root-centered causal graph;

- evaluate whether the generated graph helps verify predicted root causes in a faithful and structurally consistent way.

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
