# OpenReview forum: "Group Project Proposal$"
_tsinghua.edu.cn/THU/2026/Spring/ANM — THU 2026 Spring ANM Submission_

### Official Review · Reviewer_2zD9 · 2026-05-16

**Rating:** 7
**Confidence:** 4

**Summary:**

This proposal presents a framework for verifiable root cause analysis (RCA) in microservice systems. The main idea is to couple root cause localization with temporal evidence learning and root-centered partial causal graphs. Instead of only producing a ranked list of suspicious services or metrics, the proposed method aims to evaluate each candidate root by both its local temporal evidence and its ability to explain downstream abnormal signals through a structurally and temporally plausible partial causal graph. The proposal plans to evaluate the method on the RCAEval RE2 benchmark, using service-level and metric-level Acc@1/3/5, together with verification-oriented assessments of the generated causal graph.

**Strengths:**

Clear and meaningful problem motivation.
The proposal correctly points out that practical RCA requires more than localization accuracy. Operators also need explanations that show why a predicted root cause is plausible.
Good emphasis on faithful explanation.
A strong point of this project is that the explanation is not treated as a post-hoc artifact. The root-centered partial causal graph is intended to be built from the same evidence used for ranking, which improves the credibility of the explanation.
Reasonable use of temporal evidence.
The proposal identifies useful RCA signals such as anomaly strength, onset, persistence, magnitude, and temporal order. These are important for distinguishing true root causes from downstream symptoms.
Practical benchmark choice.
Using RCAEval RE2 is appropriate because it provides a relevant benchmark setting for microservice RCA and allows comparison with existing RCA methods.
Good balance between learning and structure.
The method does not rely purely on black-box learned representations. It also uses service dependency structure as a constrained prior, which is suitable for microservice systems.

**Weaknesses:**

The method is still described at a high level.
The proposal explains the main idea well, but many algorithmic details are missing. For example, it is not clear how temporal evidence scores are computed, how they are trained, or how they are integrated into the final ranking function.
The construction of the partial causal graph needs more detail.
The proposal says that the graph should be root-centered and structurally/temporally plausible, but it does not specify the exact graph generation algorithm, edge scoring method, pruning strategy, or how temporal lags are modeled.
Evaluation of graph quality may be difficult.
The proposal includes verification-oriented assessment, but it is unclear whether the benchmark provides ground-truth causal graphs or only root labels. Without reliable graph-level labels, evaluating the quality of the generated causal graph may become subjective or indirect.
Potential risk of circular explanation.
Since the same temporal evidence is used for both ranking and explanation, the explanation may be faithful to the model but not necessarily causally correct. The proposal should clarify how it will avoid producing a graph that only justifies the model’s score rather than reflecting real propagation.
Baselines and ablation studies are not clearly specified.
The proposal should include more concrete baselines, such as statistical RCA methods, learned RCA methods, topology-based methods, and variants without temporal evidence or without graph support. This would make it easier to isolate the contribution of each component.

**Questions:**

How exactly will temporal evidence such as onset, persistence, and anomaly strength be computed or learned? Are these supervised, self-supervised, or rule-based signals?
How will the root-centered partial causal graph be constructed? Will edges be selected from the service dependency graph, learned from telemetry, or inferred from temporal ordering?
Does RCAEval RE2 provide any ground truth for propagation paths or causal graphs? If not, how will the quality of the generated partial causal graph be evaluated objectively?
How will the final candidate score combine local evidence and graph support? Is there a formal scoring function?
What baselines will be used for comparison, and what ablation studies will be conducted to show the individual value of temporal evidence learning, service-structure constraints, and partial causal graph support?
How will the method handle cases where the true root cause has weak local anomaly evidence but strong downstream effects?

---

### Official Review · Reviewer_BzQr · 2026-05-17

**Rating:** 10
**Confidence:** 4

**Summary:**

This framework identifies the root cause of microservice failures by learning temporal evidence like anomaly onset, persistence, and magnitude. It incorporates service topology as a "prior" to ensure that predicted failure paths are structurally realistic rather than just random correlations. The core output is a root-centered partial causal graph that allows system operators to visually verify the findings.

**Strengths:**

The explanations are "faithful" because the causal graph is built using the same evidence the model uses for ranking. It moves RCA from a simple "ranked list" to a verifiable explanation, which helps close the "trust gap" for operators. By using topological constraints, it avoids common pitfalls where models suggest causal links that are physically impossible in the network.

**Weaknesses:**

During real incidents, the strongest abnormal signals are often downstream symptoms rather than the root cause, which can bias the model's ranking. Errors at the service-level scoring can easily propagate down. Measuring the quality of a generated causal graph is technically difficult and requires complex new metrics.

---

### Official Review · Reviewer_eM1K · 2026-05-18

**Rating:** 8
**Confidence:** 4

**Summary:**

This proposal argues that localization alone in root cause analysis (RCA) in microservice systems is insufficient. Instead, the authors propose a framework where a predicted root cause is evaluated not only by its local anomaly evidence but also by the quality of a root-centered partial causal graph that explains how the candidate propagates to downstream abnormalities. The methodology includes learning explicit temporal evidence (anomaly strength, onset, persistence) and generating explanations directly from the same evidence used in the ranking process. Evaluation is planned on the RCAEval RE2 benchmark.

**Strengths:**

The proposal identifies a practical limitation of current RCA methods. Even when rankings are accurate, operators lack verifiable explanations. Using a root-centered partial causal graph as both a ranking criterion and an explanation is novel idea. The proposal is well-documented with recent benchmark literature. The evaluation plan appropriately includes both localization metrics and verification-oriented assessments.

**Weaknesses:**

The proposal does not say how the partial causal graph is constructed.

**Questions:**

1.	Can you give more details about how the root-centered partial causal graph is generated from the model?

---

### Official Review · Reviewer_2s8C · 2026-05-18

**Rating:** 4
**Confidence:** 5

**Summary:**

[AI Review] This paper proposes 'Verifiable RCA,' a root cause analysis method using temporal evidence and root-centered partial causal graphs. However, the submission is extremely preliminary and reads as an idea sketch rather than a technical proposal. It lacks any technical content—zero equations, zero algorithms, and zero architecture diagrams. The core novelty claim (a graph-feedback mechanism) is never formalized, and the evaluation methodology suffers from circular reasoning: evaluating graph quality requires knowing the root cause in advance. The proposal does not differentiate itself from prior graph-based RCA methods like MicroCause and CausalRCA, and the proposed 'temporal evidence learning' appears to replicate existing statistical methods such as BARO features. The paper has only 5 references, an empty progress section, no baselines, no ablation plan, and no timeline. Score: 3/10.

**Strengths:**

1. The problem of verifiable root cause analysis in microservice architectures is relevant and important.
2. The idea of combining temporal evidence with partial causal graphs for RCA has some conceptual merit.
3. Identifying verification and interpretability as goals for RCA systems addresses real practical needs.

**Weaknesses:**

1. Zero technical content: no equations, no algorithms, no architecture diagrams—the submission is an idea sketch, not a research proposal.
2. The core novelty claim (graph-feedback mechanism) is entirely undefined and never formalized.
3. Circular reasoning in evaluation: measuring graph quality requires already knowing the root cause.
4. No differentiation from prior work such as MicroCause and CausalRCA, which already perform graph-based RCA.
5. Proposed 'temporal evidence learning' replicates existing statistical methods (BARO features) without acknowledgment.
6. Phantom reference [4] (TCN paper) cited but never discussed in text.
7. Only 5 references in a field with dozens of relevant papers—insufficient literature review.
8. Empty 'Progress So Far' section with no evidence of any work completed.
9. Vague verification metrics with hedging language and no concrete definitions.
10. No baselines, no ablation study plan, and no project timeline provided.

**Questions:**

1. How exactly does the proposed graph-feedback mechanism work? Please provide a formal definition or mathematical formulation.
2. How do you resolve the circular dependency where evaluating graph quality requires knowing the root cause a priori?
3. What specifically differentiates this approach from MicroCause and CausalRCA?
4. Can you provide the scoring function, model architecture, and loss function details?
5. What is the concrete timeline and what baselines will be used for evaluation?

---

### Official Review · Reviewer_cCcU · 2026-05-18

**Rating:** 5
**Confidence:** 3

**Summary:**

The authors propose an intriguing conceptual shift in microservice root cause analysis: instead of generating causal graphs merely as post-hoc explanations, the model should evaluate root candidates based on the quality of the "root-centered partial causal graph" they induce. While the motivation is highly relevant to practical operations, the proposal currently reads as a high-level position paper rather than a concrete technical plan.

**Strengths:**

1.Strong Practical Motivation: The proposal correctly identifies a major pain point in microservice operations: operators need interpretable, verifiable evidence to trust a model's predictions during cascading failures, not just a black-box ranked list.
2.Conceptual Innovation: The idea of elevating the partial causal graph from a "descriptive artifact" to a core structural constraint used during the evaluation of a root candidate is theoretically appealing.
3.Clear Evaluation Scope: The proposal identifies a specific, modern benchmark dataset (RCAEval 'RE2') and defines both standard localization metrics and custom verification-oriented metrics.

**Weaknesses:**

1.Missing Baselines: The evaluation plan lists primary metrics  but fails to name a single baseline model against which this new approach will be compared.
2.Undefined Mechanisms: It is unclear how the "temporal evidence learning"  will be implemented. The proposal mentions extracting anomaly strength, onset, and persistence, but does not specify whether this uses statistical heuristics, temporal convolutions, or recurrent networks.

**Questions:**

1.How is the "quality" of the partial causal graph mathematically formulated, and how does it integrate into the loss function or scoring mechanism for candidate ranking?
2.What specific machine learning architectures (e.g., GNNs, Transformers, statistical methods) will be used to extract the explicit temporal evidence?
3.Which specific state-of-the-art RCA baselines will you use to demonstrate the superiority of your verifiable approach on the RCAEval benchmark?
4.How do you plan to measure whether masking graph-supported evidence changes the predicted root score  without falling into circular reasoning?